# Inflammation Associated Pancreatic Tumorigenesis: Upregulation of Succinate Dehydrogenase (Subunit B) Reduces Cell Growth of Pancreatic Ductal Epithelial Cells

**DOI:** 10.3390/cancers12010042

**Published:** 2019-12-21

**Authors:** Sascha Rahn, Philippe Dänzer Barbosa, Julia Luisa Möller, Nourhane Ammar, Tobias Demetrowitsch, Ole Helm, Daniela Wesch, Bence Sipos, Christoph Röcken, Karin Schwarz, Heiner Schäfer, Susanne Sebens

**Affiliations:** 1Institute for Experimental Cancer Research, University of Kiel and University Medical Center Schleswig-Holstein (UKSH) Campus Kiel, 24105 Kiel, Germany; sascha.rahn@email.uni-kiel.de (S.R.); nourhane.h.ammar@gmail.com (N.A.); hschaef@1med.uni-kiel.de (H.S.); 2Institute of Biochemistry, University of Kiel, 24118 Kiel, Germany; 3Institute for Experimental Hematology, Hannover Medical School, 30625 Hannover, Germany; DaenzerBarbosa.Philippe@mh-hannover.de; 4Fraunhofer Institute for Toxicology and Experimental Medicine, 30625 Hannover, Germany; 5Department of Hematology and Oncology, University Medical Center Schleswig-Holstein (UKSH) Campus Kiel, 24105 Kiel, Germany; luisafaltinek@gmail.com; 6Institute for Human Nutrition & Food Science, Department of Food Technology, University of Kiel, 24118 Kiel, Germany; tdemetrowitsch@foodtech.uni-kiel.de (T.D.); kschwarz-2@foodtech.uni-kiel.de (K.S.); 7Institute of Immunology, University of Kiel and UKSH Campus Kiel, 24105 Kiel, Germany; daniela.wesch@uksh.de; 8BAG für Pathologie und Molekularpathologie Stuttgart, 70176 Stuttgart, Germany; Bence.Sipos@med.uni-tuebingen.de; 9Department of Pathology, UKSH Campus Kiel, 24105 Kiel, Germany; christoph.roecken@uksh.de

**Keywords:** pancreatic cancer, SDHB, macrophages, inflammatory stroma, chronic pancreatitis

## Abstract

Pancreatic ductal adenocarcinoma (PDAC) is amongst the most fatal malignancies and its development is highly associated with inflammatory processes such as chronic pancreatitis (CP). Since the succinate dehydrogenase subunit B (SDHB) is regarded as tumor suppressor that is lost during cancer development, this study investigated the impact of M1-macrophages as part of the inflammatory microenvironment on the expression as well as function of SDHB in benign and premalignant pancreatic ductal epithelial cells (PDECs). Immunohistochemical analyses on pancreatic tissue sections from CP patients and control individuals revealed a stronger SDHB expression in ducts of CP tissues being associated with a greater abundance of macrophages compared to ducts in control tissues. Accordingly, indirect co-culture with M1-macrophages led to clearly elevated SDHB expression and SDH activity in benign H6c7-pBp and premalignant H6c7-kras PDECs. While siRNA-mediated SDHB knockdown in these cells did not affect glucose and lactate uptake after co-culture, SDHB knockdown significantly promoted PDEC growth which was associated with increased proliferation and decreased effector caspase activity particularly in co-cultured PDECs. Overall, these data indicate that SDHB expression and SDH activity are increased in PDECs when exposed to pro-inflammatory macrophages as a counterregulatory mechanism to prevent excessive PDEC growth triggered by the inflammatory environment.

## 1. Introduction

Pancreatic ductal adenocarcinoma (PDAC) is the 4th most frequent cause of cancer-related deaths in Western countries [1,2]. The overall 5-year-survival rate is still below 10% which is mainly due to the lack of reliable tests for early detection of this tumor. Therefore, PDAC is commonly diagnosed in an advanced incurable stage [3,4]. Besides age and life style factors such as smoking, a long-standing chronic pancreatitis (CP) has been shown to be an important risk factor for PDAC development [5,6,7]. Different precursor lesions have been identified which can give rise to PDAC with pancreatic intraepithelial neoplasia (PanIN) being the most frequent and best characterized precursor lesions [3,4]. PanINs are also abundant in CP and like CP PDAC is characterized by a profound inflammatory stroma accounting for up to 90% of the tumor mass. This so-called desmoplastic reaction already starts to evolve at the stage of precursor lesions or CP [5,8,9,10]. In addition to a large non-cellular compartment, CP and PDAC stroma comprise various non-neoplastic cell populations including myofibroblasts, T cells and macrophages [8,9,10]. Among these, macrophages constitute one of the earliest and major immune cell populations in both CP and PDAC stroma [8,9,10]. Functionally, macrophages are roughly divided into classically activated or pro-inflammatory M1-macrophages and alternatively activated or anti-inflammatory M2-macrophages [11]. The M1-polarization can be induced by Granulocyte/Macrophage-Colony Stimulating Factor (GM-CSF) or bacterial products like lipopolysaccharides (LPS) and is characterized by an increased secretion of pro-inflammatory cytokines like IL-1β, IL-6 and TNF-α [12]. The fact that M1-macrophages are more abundant in CP than PDAC stroma is constituent with the role of this phenotype in initiation, promotion and maintenance of inflammatory responses [13,14]. Based on their gene expression profile and secretome, (tumor-associated) macrophages are assumed to essentially contribute to malignant transformation as well as tumor growth, apoptosis resistance and cell invasion [15].

Besides the central role of inflammatory cells such as macrophages in PDAC initiation and progression, reprogramming of the cellular metabolism in pancreatic ductal epithelial cells (PDECs) might also contribute to increased proliferative activity and resistance towards apoptotic stimuli in PDECs, thereby facilitating malignant progression [16]. Indeed, recent studies indicate a high addiction of PDAC progression to metabolic alterations and tumor metabolism is assumed to be its “Achilles Heel” [16,17]. As a hallmark of cancer, metabolic reprogramming has been shown to essentially promote tumor development [18,19]. For the first time, Otto Warburg described that tumor cells show an altered metabolism that is characterized by high glycolytic activity and lowered oxidative phosphorylation [20]. In this context, tumor cells may fuel glycolysis-associated pathways to metabolize glucose and generate building blocks for enhanced DNA synthesis and redox homeostasis, e.g., the pentose phosphate pathway (PPP), or adapt their metabolism for utilization of alternative carbon sources like glutamine [21,22]. Importantly, the metabolic status of tumors and putatively also their precursor cells seems to be highly dynamic and depends on microenvironmental conditions (e.g., stromal composition due to inflammatory processes) [23,24,25,26,27,28]. Succinate dehydrogenase (SDH) is an important mitochondrial enzyme involved in the tricarboxylic acid (TCA) cycle as well as the electron transport chain. The SDH subunit B (SDHB) being one of the four subunits of SDH and the essential one for electron transport is regarded as tumor suppressor because its mutational inactivation or its decreased expression has been observed in several tumor entities [29,30]. In certain tumor entities such as pheochromocytomas genetic predisposition together with SDHB mutations have been described to lead to functional loss of SDHB [31,32]. Thus, regulation of SDHB expression might be a critical determinant in controlling cell growth and behavior of PDECs in premalignant pancreatic lesions thereby counteracting accelerated cell proliferation and malignant transformation. 

Addressing this issue, the present study investigated the impact of an M1-macrophage shaped inflammatory microenvironment on the expression of SDHB and SDH activity in benign and premalignant PDECs in order to get a better understanding of the role of SDHB in inflammation-associated pancreatic carcinogenesis. 

## 2. Results

### 2.1. Elevated SDHB Expression in Ducts of Chronically Inflamed Pancreata

Since CP represents one major risk factor for PDAC development [5,6,7], it was investigated whether a chronically inflamed microenvironment alters gene expression of SDHB, a metabolic key protein and described tumor suppressor, in pancreatic ducts. Thus, the expression of SDHB was examined in pancreatic tissues of CP patients and compared to pancreatic tissue sections from individuals who did not die from any pancreatic disease (control tissue). As shown in Figure 1A+B, strong SDHB expression was detectable in ducts exhibiting considerable morphological alterations associated with PanINs in inflamed areas of tissues from CP patients.

In contrast, ducts showing a physiological morphology and being located in an uninflamed microenvironment also exhibited SDHB expression but much less pronounced than under CP conditions (Figure 1A+B). Thus, in CP tissues 21.2% of ducts were characterized by strong SDHB expression, while only 3.3% in control tissues (Figure 1B). Furthermore, in 7/10 CP cases an increased SDHB expression could be correlated with an elevated presence of CD68+ macrophages (Figure 1C). Thus, as determined by staining of CD68 in a former study [14], macrophages were highly abundant in the stroma next to epithelial ducts while only rarely found in healthy control tissues (Figure 1B indicated by the arrow heads). Overall, these data suggest that SDHB expression is elevated in pancreatic ductal epithelium when exposed to an inflammatory microenvironment enriched with macrophages.

### 2.2. Co-culture with M1-macrophages Increases SDHB Expression in H6c7-pBp and H6c7-kras Cells

In order to investigate if paracrine interactions with pro-inflammatory macrophages being highly abundant in the stroma of CP tissues [14] lead to an increased expression of SDHB and other metabolism-associated enzymes in benign and premalignant PDECs, an indirect co-culture system was applied using benign H6c7-pBp and premalignant H6c7-kras cells and the expression of SDHB was analyzed. In line with the findings from staining of CP tissues, SDHB expression was clearly enhanced in H6c7-pBp and H6c7-kras cells after co-culture with M1-macrophages (M1-macrophages) as demonstrated by western blotting (Figure 2A) and immunofluorescence staining (Figure 2B). Interestingly, glucose-6-phosphate dehydrogenase (G6PD), the rate limiting enzyme of the PPP that catalyzes the oxidation of glucose-6-phosphate into 6-phosphoglucono-δ-lactone generating NADPH+H+, was concomitantly expressed at lower levels in PDECs from co-culture compared to those from mono-cultures (Figure 2A+B). 

In contrast, the expression of other metabolic enzymes (transketolase, transaldolase, glycerinaldehyde-3-phosphate dehydrogenase, lactate dehydrogenase A/B) and metabolite transporters (monocarboxylate transporter 1/2) in H6c7-pBp and H6c7-kras cells was not affected by co-culture conditions (data not shown). In line with the elevated SDHB expression, both PDEC lines exhibited an elevated SDH activity after co-culture with M1-macrophages, with an effect being more pronounced in H6c7-pBp than in H6c7-kras cells (Figure 2C). Accordingly, elevated SDHB expression and SDH activity seems to be associated with an enhanced oxidative phenotype of the cells (Appendix A, Figure A1). In summary, these data support the view that paracrine interaction with pro-inflammatory macrophages increases the expression of SDHB as well as SDH activity and simultaneously decreases the expression of G6PD in PDECs.

### 2.3. siRNA Mediated Downregulation of SDHB Expression Diminishes SDH Activity in H6c7-pBp and H6c7-kras cells after Mono- and Co-culture with M1-macrophages

In order to investigate the role of SDHB in PDECs under inflammatory conditions, SDHB expression was silenced by transient siRNA-mediated knockdown. Analysis of SDHB expression after siRNA treatment revealed a clearly reduced SDHB expression in both H6c7-pBp and H6c7-kras cells under mono- and co-culture conditions as demonstrated by western blotting (Figure 3A) and immunofluorescence staining (Figure 3B+C). Notably, G6PD expression was markedly elevated in siSDHB transfected PDECs in comparison to siCtrl treated cells (Figure 3B) but none of the other examined enzymes (transketolase, transaldolase, glycerinaldehyde-3-phosphate dehydrogenase, lactate dehydrogenase A/B) or transporters (monocarboxylate transporter 1/2) (data not shown). Furthermore, SDHB knockdown also reduced SDH activity except in co-cultured H6c7-pBp cells (Figure 3D). In summary, these data show that siRNA mediated SDHB silencing specifically reduces SDHB expression and activity. 

### 2.4. Impact of Paracrine Interaction with M1-macrophages and SDHB Knockdown on Uptake of Glucose and Lactate in H6c7-pBp and H6c7-kras Cells

An elevated uptake of glucose might be indicative for an increased metabolic flux into anabolic pathways that foster cell proliferation, while a reduced uptake might indicate a metabolic switch towards utilization of other carbon sources and presumably a less energy consuming state—e.g., quiescence. Thus, it was next investigated whether glucose uptake is impacted by M1-macrophages and if so, whether this is dependent on elevated SDHB expression. As shown in Figure 4A+B 2-NBDG uptake assay revealed that mono-cultured H6c7-kras cells exhibited a significantly lower glucose uptake compared to H6c7-pBp cells (Figure 4A) and that the presence of M1-macrophages led to a reduced glucose uptake in both PDEC lines compared to mono-culture. In line with the more pronounced effects of the co-culture on SDHB expression and SDH activity, glucose uptake was more reduced in H6c7-pBp cells than in H6c7-kras cells (Figure 4B). While under mono-culture, siRNA-mediated knockdown of SDHB also reduced glucose uptake compared to control transfected cells, no further reduction was observed after SDHB knockdown in the PDECs when exposed to M1-macrophages (Figure 4B). Beside glucose, lactate is an important alternative energy source, particularly used by tumor cells [33]. Hence, lactate uptake in mono- and co-cultured H6c7-pBp and H6c7-kras cells was determined. Similar to the findings for glucose uptake, lactate was also taken up less by mono-cultured H6c7-kras cells compared to H6c7-pBp cells (Figure 4C) and both PDEC lines exhibited a reduced lactate uptake after co-culture with M1-macrophages compared to mono-cultured cells (Figure 4D). Noteworthy, this effect could not be reversed by knockdown of SDHB. Overall, these findings indicate that PDECs exhibit a reduced uptake of glucose and lactate in the presence of M1-macrophages. Since the macrophage mediated effect on glucose and lactate uptake was not influenced by knockdown of SDHB, the observed increase in SDHB expression and SDH activity under these conditions might be rather the consequence than the cause of the altered glucose/lactate uptake in PDECs.

### 2.5. Elevated Cell Growth of H6c7-pBp and H6c7-kras Cells in the Presence of M1-macrophages is Further Enhanced by Downregulation of SDHB Expression 

Since an enhanced and uncontrolled proliferative activity is a characteristic property gained during malignant transformation and macrophages have been described to promote epithelial cell proliferation [24], it was investigated if M1-macrophages affect cell growth of H6c7-pBp and H6c7-kras cells and whether this relates to the elevated SDHB expression. Vital cell countings revealed slightly increased cell numbers of PDECs co-cultured with M1-macrophages in comparison to mono-cultured cells, with effects being again more pronounced in H6c7-pBp cells than in H6c7-kras cells (Figure 5A). In line with these findings, the proportion of Ki67+ cells was elevated (Figure 5B). Moreover, caspase-3/7 activity indicative for apoptotic cells was slightly increased (Figure 5C) in H6c7-pBp cells and slightly decreased in H6c7-kras cells under co-culture conditions. Importantly, siRNA-mediated silencing of SDHB expression clearly elevated the number of vital as well as the proportion of proliferating cells and reduced caspase-3/7 activity of both PDEC lines under either culture condition. Notably, this effect was more pronounced in PDECs from co-culture with M1-macrophages (Figure 5A+B). Hence, highest vital cell numbers were counted in co-cultured H6c7-pBp and H6c7-kras cells when SDHB expression was silenced (both 2.16-fold). These data suggest that elevated expression of SDHB under co-culture conditions is a counterregulatory mechanism to control excessive cell growth of PDECs in the presence of pro-inflammatory macrophages.

## 3. Discussion

CP, one of the main risk factors for PDAC containing precursor lesions such PanINs, as well as PDAC are characterized by a high abundance of macrophages [14,34]. It has been shown that macrophages promote malignancy associated alterations already in PDECs paving the way for PDAC development [7,10,13]. SDHB belongs to the SDH complex in mitochondria and has been described as a tumor suppressor [35]. Accordingly, a loss of SDHB function can be observed in many tumor entities and during PDAC progression a reduction in SDHB expression has been described, too [36,37,38,39]. Our immunohistochemical analyses of CP tissues revealed a higher SDHB expression in PanINs compared to ducts of unaffected pancreata. Interestingly, a high abundance of macrophages close to PanINs showing increased SDHB expression could be detected in the majority of CP tissues while normal ducts with lower SDHB expression were located in an unobtrusive microenvironment suggesting a link between pro-inflammatory macrophages and elevated SDHB expression in PanINs. In line with these *in situ* findings, indirect co-culture with pro-inflammatory macrophages, mimicking the paracrine interplay of macrophages and PDECs in CP tissues, led to an elevated SDHB expression and SDH activity on the one hand and a reduced expression of G6PD on the other hand in both H6c7-pBp and H6c7-kras cells. With regard to these findings, an elevated SDHB expression and SDH activity (along with a reduced expression of G6PD) may relate to a metabolic switch of the cells which is associated with enhanced OXPHOS activity contrasting the tumor associated Warburg phenotype that is characterized by an increased glucose uptake and elevated glycolysis under aerobic conditions [21,40,41]. In line with this assumption, glucose uptake was rather reduced in both PDEC lines after co-culture with M1-macrophages and similar observations were also made with respect to the uptake of lactate, an alternative carbon source especially for tumor cells [23,42,43,44]. Furthermore, first Seahorse analyses indicate that co-cultured PDECs indeed exhibit an enhanced oxygen consumption (Appendix A, Figure A1). Overall, these results suggest that PDECs adapt a more oxidative than glycolytic phenotype due to paracrine interaction with M1- macrophages. Ongoing studies have to reveal in more detail how this relates to a metabolic shift in the cells.

In order to examine whether elevated SDHB expression might be the cause or the consequence of an altered metabolism in PDECs in the presence of macrophages, SDHB expression was silenced by siRNA transfection. However, knockdown of SDHB did not reverse the co-culture mediated reduction of glucose and lactate uptake but it independently led to a diminished uptake of the two metabolites. Interestingly, SDHB silencing significantly increased cell growth due to elevated proliferation and reduced effector caspase activity in both PDEC lines under co-culture conditions. Hence, G6PD expression was downregulated in PDECs in the presence of M1-macrophages but markedly upregulated when SDHB expression was silenced in PDECs. Notably, an increased metabolic flux into the PPP due to increased expression and activity of G6PD would provide more NADPH+H+ for anabolic processes as well as ribose for DNA synthesis and, thereby, provide the basis for the enhanced proliferative activity of PDECs transfected with SDHB-targeting siRNA. Moreover, increased NADPH levels may also provide redox equivalents for protecting the cells against reactive oxygen species (ROS) [45]. In conclusion, our data indicate that PDECs adapt metabolic alterations under inflammatory conditions. These adaptations are characterized by an increased SDHB and decreased G6PD expression. On the functional level, SDHB levels affect SDH activity as well as the regulation of cell proliferation and apoptosis induction and, by that, counteract M1-macrophages promoted cell growth. In line with these findings, it has been shown in e.g., ovarian and colorectal cancer cells that a lack of SDHB leads to HIF-1-dependent metabolic alterations towards a more glycolytic phenotype [36,46] which is associated with induction of Epithelial-Mesenchymal-Transition (EMT) and elevated proliferation [37,38,47].

Since in our study an indirect co-culture setting was used, the observed alterations in PDECs have to be mediated in a paracrine fashion. Indeed, TNF-α which is highly secreted by M1-macrophages [48] has been shown to induce the expression of anti-apoptotic genes such as Bcl-2 and c-FLIP genes via activation of NF-κB, thereby leading to a protection from apoptosis induction and enhanced survival [49,50]. Moreover, TNF-α signaling is also known to block Y box binding protein 1 (YB-1)-mediated degradation of SDHB mRNA and thereby leads to its enhanced expression [51,52] which might represent an explanation for enhanced SDHB expression in PDECs in the presence of M1-macrophages. 

It has been shown that SDH dissociation into membrane bound SDHC/D and soluble SDHA/B in response to pro-apoptotic stimuli amplifies apoptosis induction by transferring electrons to oxygen and, thereby, generating harmful intracellular oxygen radicals. With regard to the tumor suppressor function of SDHB its upregulation in PDECs due to co-culture with M1-macrophages might, thus, be considered as a counterregulatory, tumor preventive mechanism in order to sensitize cells for apoptotic stimuli and, thereby, impede excessive cell proliferation under pro-inflammatory conditions that favor malignant transformation [35,53,54,55]. In this context, Guzy et al. provided evidence that SDHB does not exert its tumor suppressive properties via bioenergetic regulation [35]. In line with this, it has been shown that a lack or reduced expression of SDHB does not completely diminish succinate oxidizing activity but the electron transport from the catalytic domain in SDH subunit A (SDHA) into the electron transport chain. Thereby, electrons get stuck in SDHA leading to a Flavin adenine dinucleotide (FAD) auto-oxidation and the generation of highly reactive superoxide. These ROS subsequently activate HIF-1-mediated signaling which in turn enhances proliferation and vascularization [36,46]. Furthermore, ROS also promote genetic instability, thereby, paving the way for tumor promoting mutations and malignant transformation [53]. 

## 4. Materials and Methods 

### 4.1. Immunohistochemical Staining of SDHB in Human Pancreatic Tissues

Expression of SDHB in pancreatic ducts was detected in 3 µm-thick formalin-fixed paraffin-embedded tissue sections of patients suffering from CP and patients without putatively pathological pancreatic diseases. Tissue sections were kindly provided by B. Sipos (BAG für Pathologie und Molekularpathologie Stuttgart, Germany). In case of the CP tissue sections, informed consent was obtained and research was approved by the ethics committee of the University Hospital Tübingen (reference number: 470/210BO1). Research on the pancreatic tissue sections of non-pathological origin was authorized by the ethics committee of the Semmelweis University Budapest (reference number: 140-1/1996) and the need for an informed consent was waived. First, deparaffinization of tissue sections was performed by incubating sections two-times in xylene for 10 min. Afterwards, samples were rehydrated applying a descending alcohol series, simultaneously blocking endogenous peroxidases by adding 1.5% (v/v) H_2_O_2_. Then, tissue sections were washed for 10 min with PBS before antigen retrieval was performed by incubating the sections in a steamer for 20 min in pre-warmed antigen retrieval buffer (citrate buffer, pH 6.0). After cooling down to room temperature (RT), tissues were washed with PBS and unspecific binding-sites were blocked by incubation in PBS supplemented with 0.3% (v/v) Triton X-100 (PBS-T) and 4% (w/v) BSA for 1 h at RT. Then, anti-SDHB primary antibody (clone EPR10880, Abcam, Cambridge, UK) diluted at 2.8 µg/mL in PBS-T supplemented with 1% (w/v) BSA was applied and tissue sections were incubated in a wet chamber overnight at 4 °C. Detection was carried out by EnVision + system-HRP labeled polymer anti-mouse or anti-rabbit, followed by administration of AEC Substrate (both Dako Diagnostika, Hamburg, Germany). Nuclear staining was performed with Mayer’s Haemalaun (AppliChem, Darmstadt, Germany). Respective isotype controls were used to verify staining specificity. Semiquantitative evaluation of SDHB expression was performed using the following scoring system: 1= weak staining intensity, 2= intermediate staining intensity, 3= strong staining intensity. CD68 staining was already performed in a former study [14]. Images were taken with a Lionheart FX Automated microscope making use of the software Gen5 (both from BioTek, Bad Friedrichshall, Germany). 

### 4.2. Pancreatic Ductal Epithelial Cell Lines

The cell lines H6c7eR-pBp (harboring no PDAC-related mutations and being non-tumorigenic) and H6c7eR-kras (harboring k-Ras G12V mutation and being partially tumorigenic), used as models for benign and premalignant human PDECs present in CP tissues, respectively, were kindly provided by M.S. Tsao (Ontario Cancer Institute, Toronto, ON, Canada) and cultured as described previously [56,57,58]. All cell lines have recently been authenticated by STR-analysis.

### 4.3. Isolation and Generation of Pro-inflammatory M1-like Polarized Macrophages

Macrophages were generated from monocytes of healthy donors. Informed consent was received from all donors. Monocytes were obtained from lymphocyte retaining systems from blood donations at the Institute for Transfusion Medicine (Kiel, Germany). They were isolated by density gradient centrifugation followed by counterflow centrifugation and differentiated into pro-inflammatory M1-macrophages via stimulation with 50 ng/mL GM-CSF (240 U/mL) (BioLegend, Fell, Germany) for seven days. The phenotype of M1-macrophages was routinely validated as described previously [13,28].

### 4.4. Indirect Co-cultures of PDECs and M1-macrophages

PDECs were seeded at 1.5 × 10^4^ cells/well in 1 mL/well HPDE medium (50% (v/v) RPMI-1640 supplemented with 10% (v/v) fetal calf serum, 2 mM L-glutamine and 50% (v/v) keratinocyte serum-free medium supplemented with 50 µg/mL bovine pituitary extract, 5 ng/mL Epidermal Growth Factor and 4 µg/mL puromycin) into a 12-well plate. In parallel, M1-macrophages were seeded at 1.5 × 10^5^ cell/well in 1 mL/well RPMI-1640 medium supplemented with 10% fetal calf serum and 2 mM L-glutamine into a 12-well plate transwell insert (0.4 µm pore size, Greiner, Frickenhausen, Germany). After 24 hours, media were aspirated and replaced by RPMI-1640 medium supplemented with 10% fetal calf serum and 2 mM L-glutamine. Finally, transwell inserts were placed into the wells containing adherent PDECs. Co-cultures were carried out for five days until subsequent analyses were performed.

### 4.5. siRNA-mediated Knockdown of SDHB Expression

In order to perform siRNA-mediated knockdown of SDHB gene expression, PDECs were seeded and cultured as described one day before harvesting M1-macrophages. Transfection of PDECs was performed 24 h after seeding. For the transfection of cells in one 12-well, 100 µl Opti-MEM serum reduced medium (Thermo Scientific, Schwerte, Germany) were mixed with 6 µL HiPerfect Transfection Reagent (Qiagen, Hilden, Germany) and 1 µL of either 10 µM Control siRNA (sc-37007) or 10 µM SDHB siRNA (sc-44088, both from Santa Cruz Biotechnology, Heidelberg, Germany) and incubated for 10 min at RT before being added to the cells. Eight hours later, medium was replaced by fresh medium.

### 4.6. Immunofluorescence Staining of SDHB and Ki67

Cells were seeded on cover slips (ø 18 mm; Thermo Scientific) and cultured as described. Immunofluorescence staining was performed as previously described [39]. The following antibodies were used: 1.7 µg/mL anti-SDHB (clone EPR10880, Abcam), 10 µg/mL anti-Ki67 clone B56; BD Biosciences, Heidelberg, Germany) and 4 µg/mL of either anti-mouse IgG-Alexa Fluor 488 or anti-rabbit IgG-Alexa Fluor546 (Thermo Scientific). Nuclear staining was performed by incubation in Hoechst 33258 (Sigma-Aldrich, St. Louis, MO, USA) diluted at 2 µg/mL in PBS for 1 hour at RT. Afterwards, samples were mounted in FluorSave Reagent (Merck Millipore, Darmstadt, Germany) on microscope slides. Staining specificity was verified by staining with isotype control antibodies (rabbit IgG, clone SP137, Abcam). Images were taken with a Keyence BZ-9000 fluorescence microscope and relative quantification was performed making use of the Hybrid Cell Count tool provided by BZ-9000 Image Analysis Application (both from Keyence, Neu-Isenburg, Germany).

### 4.7. Western Blotting

Whole cell lysates were prepared and electrophoresis as well as western blotting were performed as described [59]. The following antibodies were used: anti-mouse IgG-HRP (#7076), anti-rabbit IgG-HRP (#7074) (all from Cell Signaling, Frankfurt, Germany), anti-G6PD [H-160], anti-HSP90α/β [H-114] and anti-SDHB [FL-280] (all from Santa Cruz Biotechnology, Heidelberg, Germany). All antibodies were diluted according to the manufacturer’s instructions.

### 4.8. Determination of vital cell numbers

Cells were detached with Trypsin-EDTA (PAA, Pasching, Germany) and counted using a Neubauer counting chamber. In order to discriminate vital and dead cells a staining with trypan blue (Sigma-Aldrich, Munich, Germany) was performed. 

### 4.9. Measurement of Caspase-3/7 Activity

Caspase-3/7 activity was measured making use of the Caspase-Glo® assay (Promega, Mannheim, Germany) according to the manufacturer’s instructions and as described [59]. Samples were measured in duplicates and resulting values were normalized to the respective protein concentration.

### 4.10. Glucose Uptake Assay 

After co-culture PDECs were washed with PBS. Subsequently, cells were incubated with 500 μL co-culture medium supplemented with 10 µM 2-(N-(7-nitrobenz-2-oxa-1.3-diazol-4-yl)-amino)-2-deoxyglucose (2-NBDG, Thermo Scientific) at 37 °C for 30 min. Then, 2-NBDG supplemented medium was withdrawn and cells were washed with PBS. Afterwards, cells were detached using Trypsin/EDTA, transferred into 1.5 mL reaction tubes and 500 μL co-culture medium were added. Then, cells were centrifuged for 5 min at 500⋅g and RT. Finally, cells were resuspended in 100 μL PBS, transferred into FACS tubes and 2-NBDG uptake analysis was performed using a FACScalibur flow cytometer (Beckton Dickinson, Heidelberg, Germany).

### 4.11. Lactate uptake assay 

Lactate uptake by PDECs was examined using uniformly labeled ^14^C-lactate (Hartmann Analytics, Braunschweig, Germany). After removal of the medium, cells were equilibrated in 250 µL 10 mM Hepes/pH 7.50, 5 mM KCl, 100 mM NaCl, 1 mM MgCl_2_ (uptake buffer) containing 2 µCi (0.5 µM) ^14^C-L-lactate alone or together with 10 mM unlabeled lactate and cells were incubated for 2 h at 37 °C. Afterwards, cells were washed thrice with ice-cold PBS and then lysed in 500 µL uptake-buffer with 2% (w/v) SDS. Lysates were submitted to liquid-scintillation beta-counting (Packard TRI-CARB 2000TR Liquid Scintillation Analyzer, GMI, Ramsey, MN, USA). In parallel, protein concentrations in lysates from unlabeled cells were measured (DCTM-Assay) and used for normalization of beta-counting rates (triplicate measurements). Values of control samples (incubation with non-labeled lactate), which accounted for unspecific extracellular binding, were subtracted from corresponding sample values which were incubated without additional non-labeled lactate.

### 4.12. Determination of Succinate Dehydrogenase Activity

For determination of SDH activity in PDECs, a colorimetric assay was used (adapted to [60]), which is based on the reduction of artificial electron acceptor 2,6-dichlorphenol-indophenol (DCIP). For this purpose, transwells with macrophages were removed after co-culture. PDECs were washed with PBS, detached with Trypsin/EDTA for 10 min, transferred into a 1.5 mL reaction tube and centrifuged at 500⋅g for 5 min at 4 °C. Afterwards, cells were resuspended in 100 μL isolation buffer (210 mM mannitol, 70 mM sucrose, 10 mM 3-(N-morpholino)propanesulfonic acid, pH 7.2) and 5 μL of the cell suspension were transferred into a 96-well plate in duplicates. Afterwards, 90 μL SDH assay buffer (20 mM sodium succinate, 10 mM sodium azide, 25 mM disodium hydrogenphosphate, 25 mM potassium dihydrogenphosphate, pH 7.4) were added and incubated for 5 min at RT to allow succinate uptake and complex IV inhibition. Colorimetric reaction was started by adding 50 µM DCIP. In order to determine molar enzymatic turnover, a DCIP standard was prepared with concentrations of 0, 12.5, 25, and 50 μM DCIP in SDH Assay buffer. DCIP reduction was monitored by measuring absorbance at *λ* = 600 nm every minute for 30 min, using an Infinite® M200 Pro microplate reader (TECAN, Crailsheim, Germany). Resulting DCIP turnover was normalized to whole cell lysate protein concentration of respective samples, generating values in the dimension nmol/min/mg protein. SDH-dependent DCIP reduction was confirmed by performing dose-dependent inhibition using the specific competitive SDH inhibitor malonate in concentrations ranging from 0 to 500 mM. For this purpose, malonate was added prior to DCIP addition and the assay was performed as described.

### 4.13. Seahorse Analysis

Oxygen consumption rate as an indicator for mitochondrial oxidative phosphorylation activity in PDECs from mono- and co-culture settings was examined with a Seahorse XF Analyzer performing a XFp Mito Stress Test (both Agilent, Santa Clara, CA, USA) according to the manufacturer’s instructions. In preparation for the measurement, XFp Extracellular Flux Cassettes were incubated with 200 µL Calibrant Solution over night at 37 °C and XFp Cell Culture Miniplates were coated with 0.01% poly-D-lysine. Prior to the measurement, 4 × 10^4^ PDECs were washed twice in Mito-Medium, resuspended in 175 µL Mito-Medium and transferred to the prepared XFp Cell Culture Miniplate. Afterwards, cells were incubated for 1 h at 37 °C, 5% CO_2_ and 85% relative humidity. Injection ports of the device were filled with 1 µM oligomycin, 0.5 M FCCP and 0.5 M rotenone + antimycin A, respectively. Finally, XFp Extracellular Flux Cassette was equilibrated in the Seahorse XF Analyzer, XFp Cell Culture Minplate was loaded and XFp Mito Stress Test started. Data evaluation was performed with the Wave Desktop V2.2 software (Agilent, Santa Clara, CA, USA).

### 4.14. Statistical Analysis

Statistical analyses were performed using SigmaPlot v12.5 provided by Systat (Erkrath, Germany). First, data were tested for normality and equal variance by Shapiro-Wilk and Equal Variance test, respectively. For comparison of two-groups comprising parametrically distributed datasets, t-test was applied. Two groups of datasets which failed normality or equal variance test were analyzed with Mann-Whitney Rank Sum test. Parametric data of multiple groups were checked with one-way analysis of variance (one-way ANOVA) for statistical significance. Non-parametrical datasets of multiple groups were analyzed with Kruskal-Wallis one-way ANOVA on ranks test. Statistically significant differences between the groups were assumed at *p*-values < 0.05 according to Student-Newman-Keuls method (parametric data) and Dunn’s method (non-parametric data), respectively. Statistically significant differences with *p*-values < 0.05 were marked with an asterisk (*).

## 5. Conclusions

Overall, this study provides experimental evidence that upregulation of SDHB expression and thereby enhanced SDH activity in PDECs under inflammatory conditions can be regarded as a control mechanism to prevent excessive PDEC cell growth and expansion supporting its role as tumor suppressor. Accordingly, reduction or loss of SDHB expression due to mutations or stroma alterations given by accumulation of inflammatory cells and released inflammatory mediators leads to deficient cell growth control in PDECs and later on PDAC cells thereby contributing to expansion of transformed cells. Future studies have to unravel in more detail how this altered SDHB expression and activity is related to an altered flux and usage of metabolites in PDECs. In fact, metabolic reprogramming as well as SDH inhibition have also been shown to essentially contribute to EMT and with this to the acquisition of an invasive and cancer stem cell phenotype [47,61] Giving the fact that macrophages represent a predominant cell population in the stroma of CP and PDAC and are known to promote EMT and cell invasion of PDECs [13], it can be speculated that they might also impact these processes via paracrine modulation of SDHB expression. Thus, loss of SDHB expression may promote tumor development and progression not only via the loss of growth control but also by conferring tumor initiating and dissemination abilities.

## Figures and Tables

**Figure 1 cancers-12-00042-f001:**
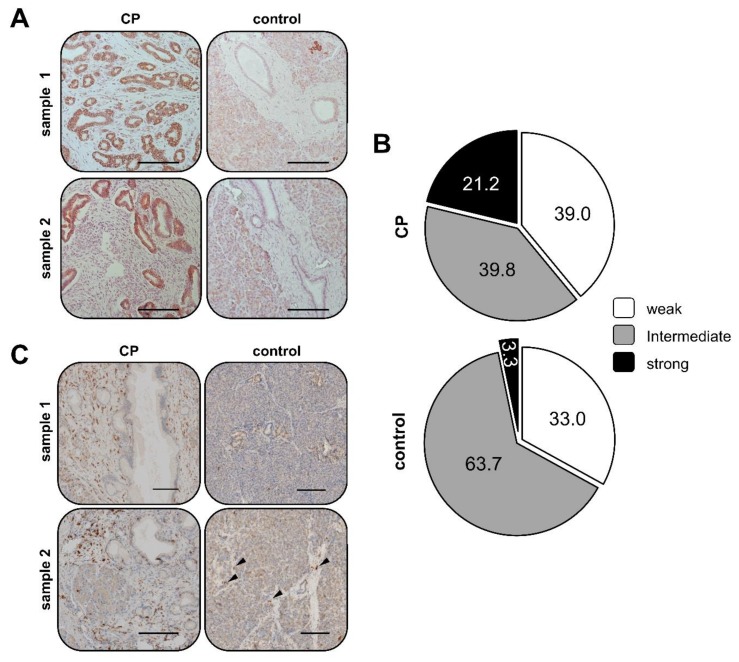
Elevated SDHB expression in pancreatic ducts of CP patients compared to ducts of healthy individuals. Representative images of immunohistochemical stainings of (**A+B**) succinate dehydrogenase subunit B (SDHB) and (**C**) CD68 in tissue sections from two CP patients (CP) and two individuals who did not die due to a pancreatic disease (control). Arrow heads indicate CD68+ cells in control samples. Scale bar = 150 µm. (**B**) Quantification of immunohistochemical analyses of SDHB staining in pancreatic ducts of CP patients (*n* = 10) and control individuals (*n* = 22). Pie charts present the proportion of ducts showing weak, intermediate and strong SDHB staining in 523 ducts of CP tissues and 739 ducts of control tissues.

**Figure 2 cancers-12-00042-f002:**
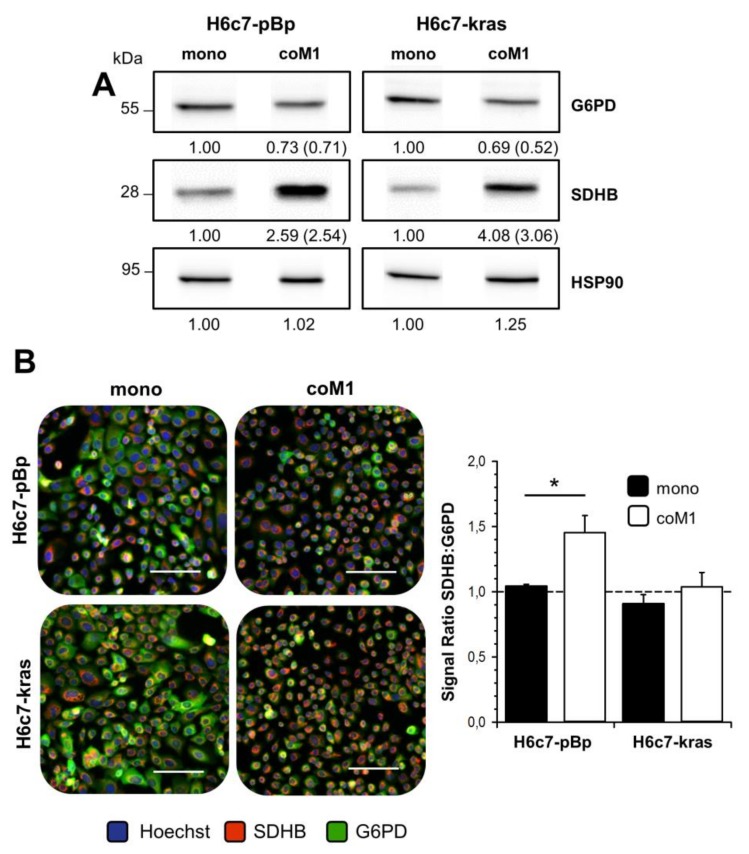
Elevated SDHB expression and SDH activity in H6c7-pBp and H6c7-kras cells after exposure to pro-inflammatory macrophages. PDECs were mono- or indirectly co-cultured with human M1-macrophages (coM1) for five days. (**A**) Representative western blots of SDHB and G6PD in whole-cell lysates from differently cultured H6c7-pBp and H6c7-kras cells. HSP90 was detected as loading control. Values from densitometric analysis are noted below the bands. Values of mono-cultured cells were set as 1. In brackets, normalized values relative to respective Hsp90 values are shown. Representative results from three independent experiments are shown. G6PD: glucose-6-phosphate dehydrogenase; SDHB: succinate dehydrogenase subunit B. (**B**) Representative fluorescence microscopic images of SDHB and G6PD staining in differentially cultured H6c7-pBp and H6c7-kras cells. Nuclear staining was performed with Hoechst. Scale bar = 100 µm. Quantification of SDHB and G6PD staining in H6c7-pBp and H6c7-kras cells (right). Bar chart presents mean values +/- SEM of 3 independent experiments. (**C**) SDH activity of mono- and co-cultured H6c7-pBp and H6c7-kras cells. Data are expressed in nmol/min SDH activity normalized to protein content (in mg) of the sample. Data represent mean +/- SEM of 3-4 independent experiments. * *p* < 0.05.

**Figure 3 cancers-12-00042-f003:**
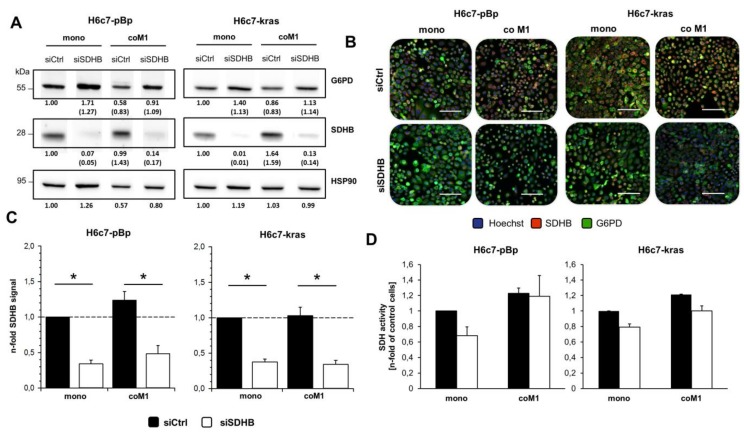
SiRNA mediated downregulation of SDHB expression diminishes SDH activity in H6c7-pBp and H6c7-kras cells after mono- and co-culture with M1-macrophages. H6c7-pBp and H6c7-kras cells were transfected with either control siRNA (siCtrl) or SDHB specific siRNA (siSDHB) followed by either mono- or indirect co-culture with human M1-macrophages (coM1) for five days. (**A**) Representative western blots of SDHB and G6PD in whole-cell lysates of differentially cultured PDECs. HSP90 was detected as loading control. Values from densitometric analysis are noted below the bands. Values of mono-cultured siCtrl treated cells were set as 1. In brackets, normalized values relative to respective Hsp90 values are shown. Representative results from 3 independent experiments are shown. G6PD: glucose-6-phosphate dehydrogenase; SDHB: succinate dehydrogenase subunit B. (**B**) Representative fluorescence microscopic images of SDHB and G6PD staining in H6c7-pBp and H6c7-kras cells after indicated treatment conditions. Nuclear staining was performed with Hoechst. Scale bar = 100 µm. (**C**) Quantification of SDHB staining in H6c7-pBp and H6c7-kras cells. Data were normalized to mono-cultured siCtrl transfected cells and are presented as mean +/- SEM of 3 independent experiments. (**D**) SDH activity of H6c7-pBp and H6c7-kras cells after indicated treatment conditions expressed as n-fold SDH activity of mono-cultured siCtrl treated cells. Data are represented as mean +/- SEM of 4 independent experiments. **p* < 0.05.

**Figure 4 cancers-12-00042-f004:**
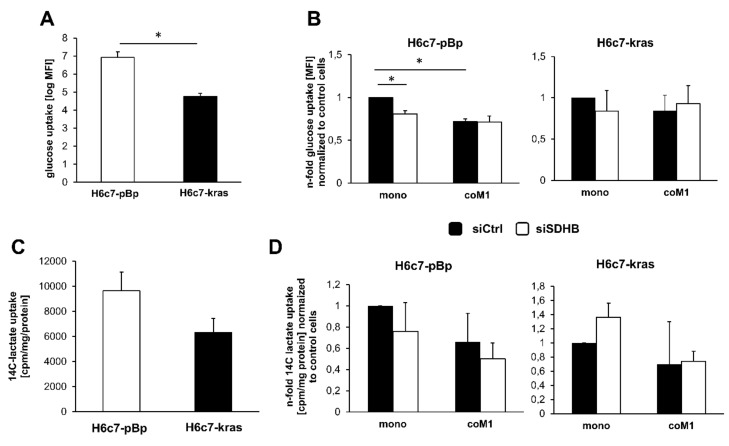
Glucose and lactate uptake in H6c7-pBp and H6c7-kras cells is reduced in the presence of M1-macrophages but not reversed by SDHB knockdown under co-culture conditions. (**A+B**) Determination of 2-NBDG uptake and (**C+D**) ^14^C lactate uptake in H6c7-pBp or H6c7-kras cells transfected with either control siRNA (siCtrl) or SDHB specific siRNA (siSDHB) followed by mono- or indirect co-culture with human M1-macrophages (coM1) for five days. Data represent mean +/- SEM of 7 independent experiments. Data are expressed as (**A**) glucose and (**C**) ^14^C-lactate uptake or as n-fold (**B**) glucose uptake and (**D**) lactate uptake normalized to respective mono-cultured siCtrl transfected cells. * *p* < 0.05.

**Figure 5 cancers-12-00042-f005:**
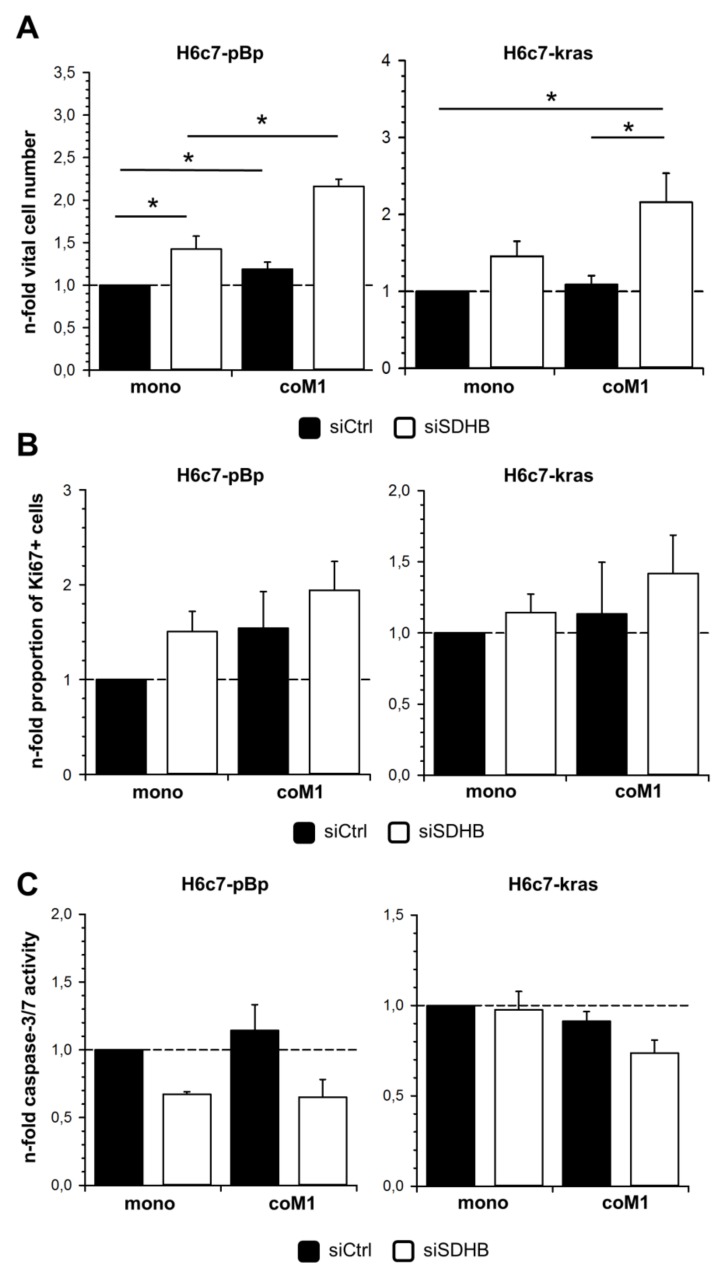
Elevated cell growth of H6c7-pBp and H6c7-kras cells in the presence of M1-macrophages is further enhanced by downregulation of SDHB expression. H6c7-pBp and H6c7-kras cells were transfected with either control siRNA (siCtrl) or SDHB specific siRNA (siSDHB) followed by either mono- or indirect co-culture with human M1-macrophages (coM1) for five days. (**A**) Determination of vital cell numbers, (**B**) proportion of Ki-67+ cells and (**C**) caspase-3/7 activity in differentially cultured cells. Data are expressed as n-fold change normalized to respective mono-cultured siCtrl transfected cells. Data represent mean +/- SEM of 3 independent experiments. * *p* < 0.05.

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
