# Peer review of "Inflammation Associated Pancreatic Tumorigenesis: Upregulation of Succinate Dehydrogenase (Subunit B) Reduces Cell Growth of Pancreatic Ductal Epithelial Cells"

_cancers, 2019, doi:10.3390/cancers12010042_

Round 1

Reviewer 1 Report

Inflammation associated initiation of pancreatic cancer: Macrophage mediated upregulation of succinate dehydrogenase (subunit B) reduces cell growth of pancreatic ductal epithelial cells under inflammatory conditions by Rahn et al.

Authors investigated the impact of an M1-macrophage inflammatory microenvironment on the expression of SDHB activity in benign and premalignant PDECs. There is not much known about the role of SDHB in inflammation-associated pancreatic carcinogenesis.

The MS is well written and logical. However, there are some concerns that need to be addressed before accepting for publication:

The characteristics of the choice of H6c7 cells for the studies is insufficiently described. Figure 2A shows increase in SDHB expression in the siControl for the Kras cells, this disagrees with results in figure 1B, please clarify. Figure 2A and B: there is discrepancy between G6PD expression in 2A and the stains in 2B with coM1, this needs clarification. C14 lactate uptake in figure 4D is too large to make any conclusion. These assays need to be repeated. If oxidative phosphorylation is suspected due to the marked decrease in glucose metabolism, OXPHOS studies (seahorse, Oroboros) is recommended. The possibility that an antioxidant response (via upregulation of PPP and NADH generation) mediates the phenotype observed with SDHB knockdown needs to be addressed and/or discussed.

Author Response

Reviewer 1:

Authors investigated the impact of an M1-macrophage inflammatory microenvironment on the expression of SDHB activity in benign and premalignant PDECs. There is not much known about the role of SDHB in inflammation-associated pancreatic carcinogenesis.

The MS is well written and logical. However, there are some concerns that need to be addressed before accepting for publication:

Request 1: The characteristics of the choice of H6c7 cells for the studies is insufficiently described.

Answer 1: In the revised version we provide more information on both cell lines (particularly in the methods section 4.2) in order to clarify the rationale for their choice.

Request 2: Figure 2A shows increase in SDHB expression in the siControl for the Kras cells, this disagrees with results in figure 1B, please clarify.

Answer 2: To our opinion the findings of figure 1B do not disagree with the results of figure 2A. In Figure 1B, we show that CP tissues contain high amounts of macrophages while in Figure 2A we demonstrate that expression levels of SDHB are increased in benign and premalignant PDECs when exposed to inflammatory macrophages. These findings correspond to the strong SDHB expression seen in epithelial ducts in CP tissues (Figure 1A+B). Importantly, we do not wish to rate whether mutant kras alone may also impact SDHB expression.

Request 3: Figure 2A and B: there is discrepancy between G6PD expression in 2A and the stains in 2B with coM1, this needs clarification.

Answer 3: For clarification, we have incorporated the values of a densitometric analysis. Thus, it is easier to identify that both PDEC lines when exposed to macrophages show a reduced G6PD expression (green staining in Figure 2B) and concomitantly an enhanced SDHB expression (red staining in Figure 2B).

Request 4: C14 lactate uptake in figure 4D is too large to make any conclusion. These assays need to be repeated.

Answer 4: The results have been presented as median and 75th percentile. Since the results did not reach statistical significance, we decided to present the results as mean and standard error of mean so that the conclusion can be appreciated easier.

Request 5: If oxidative phosphorylation is suspected due to the marked decrease in glucose metabolism, OXPHOS studies (seahorse, Oroboros) is recommended.

Answer 5: We have already started to perform seahorse analysis which supports our conclusion that macrophage mediated elevation of SDHB expression promotes an oxidative phenotype. These first data are briefly mentioned in the results section as well as the discussion and are shown in Supplementary Figure 1 at the end of the manuscript. However, since these studies are conducted in frame of a more comprehensive analysis of the metabolic phenotype of the cells and this is far beyond the scope of the current manuscript, we wish to incorporate these data in our upcoming study.

Request 6: The possibility that an antioxidant response (via upregulation of PPP and NADH generation) mediates the phenotype observed with SDHB knockdown needs to be addressed and/or discussed.

Answer 6: As requested we have addressed this issue in the discussion.

Reviewer 2 Report

Dear author,

I have read the article entitled, ''Inflammation Associated Initiation of Pancreatic Cancer: Macrophage Mediated Upregulation of Succinate Dehydrogenase (subunit B) Reduces Cell Growth of Pancreatic Ductal Epithelial Cells Under Inflammatory Conditions' with high interest. The concept of the manuscript is novel and the experiments are well carried out with proper explanation. Hence, I recommend to accept this manuscript after minor revisions.

a) The title of the manuscript is too long. It is suggested to short it for better readability.

b) Authors are requested to provide a quantification data for immunoblotting. Fig 2.A and Fig. 3A

c) The conclusion is very short and less informative. It is suggested to elaborate it little more for future outlook.

Author Response

Reviewer 2:

Dear author,

I have read the article entitled, ''Inflammation Associated Initiation of Pancreatic Cancer: Macrophage Mediated Upregulation of Succinate Dehydrogenase (subunit B) Reduces Cell Growth of Pancreatic Ductal Epithelial Cells Under Inflammatory Conditions' with high interest. The concept of the manuscript is novel and the experiments are well carried out with proper explanation. Hence, I recommend to accept this manuscript after minor revisions.

Request 1: The title of the manuscript is too long. It is suggested to short it for better readability.

Answer 1: First of all we would like to thank the reviewer for his general positive statement about our manuscript. According to the request we have shorten the title. The revised title is: “Inflammation associated pancreatic tumorigenesis: Upregulation of succinate dehydrogenase (subunit B) reduces cell growth of pancreatic ductal epithelial cells”

Request 2: Authors are requested to provide a quantification data for immunoblotting. Fig 2.A and Fig. 3A

Answer 2: As requested we have incorporated the densitometric analysis of all western blots.

Request 3: The conclusion is very short and less informative. It is suggested to elaborate it little more for future outlook.

Answer 3: We have extended the conclusion by providing an outlook on future studies but also outlining the impact of SDHB loss on other cancer properties such as invasive and tumor initiating properties.

Reviewer 3 Report

This Ms provides information about the role of SDHB and complex 2 function in the neoplastic progression of pancreatic cancer. It is of interest and worthy of publication.

I suggest a few minor amendments to the text, as follows:

Introduction

line 57: This so called desmoplastic reaction already starts to evolve already - Too many "already"! 

line 64: The M1-polarization can e.g. be induced by Granulocyte/Macrophage-Colony Stimulating Factor (GM-CSF) or bacterial products - e.g. is out of context here! 

line 67: Since this phenotype is known to initiate, foster and maintain inflammatory responses, M1-macrophages are more abundant in CP than PDAC stroma [13,14]. - The fact that macrophages are more abundant in CP is consistent with .....

lines 87-88 The SDH subunit B (SDHB) being one of the four subunits of SDH and the essential one for electron transport is regarded as tumor suppressor because its mutational inactivation or its decreased expression.... If possible the Authors should state here if there are studies demonstrating double hit inactivation of SDHB according to Knudson's model. 

Results

Lines 103-107 As shown in Figure 1A, elevated SDHB expression was detectable in ducts exhibiting considerable morphological alterations associated with PanINs in inflamed areas of tissues from CP patients. In contrast, SDHB expression was also abundant but less pronounced in ducts exhibiting a physiological morphology and being located in an uninflamed microenvironment. - Here one gets confused. First, the two observations are not necessarily "in contrast". The "abundant" (what does it mean?) expression in normal ducts in uninflamed areas is not in line with the conclusions...Could the Authors assess in a semiquantitative manner the levels of SDHB expression seen under the different conditions?

Fig 1:  For CD68 staining the low power view of controls does not allow evaluation of convincing differences

Line 163: "diametral regulation": what does it mean? This terminology is also used elsewhere, and is confusing. 

Line 183: An elevated uptake of glucose might be indicative for an increased metabolic flux into anabolic pathways that foster e.g. cell proliferation.... - Take out e.g. 

Conclusions

lines 450-456 Please explain what do you mean when you write: "Accordingly, reduction or loss of SDHB expression due to mutations or stroma mediated factors during tumor development and progression leads to deficient cell growth control in PDECs and later on PDAC cells thereby contributing to expansion of transformed cells."  What are the stoma-mediated factors? Are these the macrophages? And if so, how are the macrophages in pancreatic cancer stroma? Could well be that loss of SDHB is related to a less differentiated or EMT-prone phenotype that might be exploited in cancer, but that would also occur in differentiation and development. 

Author Response

Reviewer 3:

This Ms provides information about the role of SDHB and complex 2 function in the neoplastic progression of pancreatic cancer. It is of interest and worthy of publication.

I suggest a few minor amendments to the text, as follows:

Introduction

Request 1: line 57: This so called desmoplastic reaction already starts to evolve already - Too many "already"!

Answer 1: First of all we would like to thank the reviewer for his general positive statement about our manuscript. We have modified the text by deleting one “already”.

Request 2: line 64: The M1-polarization can e.g. be induced by Granulocyte/Macrophage-Colony Stimulating Factor (GM-CSF) or bacterial products - e.g. is out of context here!

Answer 2: We have modified the text accordingly.

Request 3: line 67: Since this phenotype is known to initiate, foster and maintain inflammatory responses, M1-macrophages are more abundant in CP than PDAC stroma [13,14]. - The fact that macrophages are more abundant in CP is consistent with .....

Answer 3: We have modified the text accordingly.

Request 4: lines 87-88 The SDH subunit B (SDHB) being one of the four subunits of SDH and the essential one for electron transport is regarded as tumor suppressor because its mutational inactivation or its decreased expression.... If possible the Authors should state here if there are studies demonstrating double hit inactivation of SDHB according to Knudson's model.

Answer 4: We have addressed this issue at the indicated position in the introduction.

Results

Request 5: Lines 103-107 As shown in Figure 1A, elevated SDHB expression was detectable in ducts exhibiting considerable morphological alterations associated with PanINs in inflamed areas of tissues from CP patients. In contrast, SDHB expression was also abundant but less pronounced in ducts exhibiting a physiological morphology and being located in an uninflamed microenvironment. - Here one gets confused. First, the two observations are not necessarily "in contrast". The "abundant" (what does it mean?) expression in normal ducts in uninflamed areas is not in line with the conclusions...Could the Authors assess in a semiquantitative manner the levels of SDHB expression seen under the different conditions?

Answer 5: In order to avoid any confusion we have modified the text. Furthermore, we have incorporated information on our semiquantitative evaluation of SDHB expression in figure 1B. The corresponding scoring system is described in the methods section 4.1.

Request 6: Fig 1: For CD68 staining the low power view of controls does not allow evaluation of convincing differences

Answer 6: To make clear that in the control tissues only single CD68+ cells are abundant, we have incorporated a statement in the results section and marked these single CD68+ cells with arrow heads in the images.

Request 7: Line 163: "diametral regulation": what does it mean? This terminology is also used elsewhere, and is confusing.

Answer 7: In order to avoid confusion and because it is not mandatory for the conclusions, we have deleted the term.

Request 8: Line 183: An elevated uptake of glucose might be indicative for an increased metabolic flux into anabolic pathways that foster e.g. cell proliferation.... - Take out e.g.

Answer 8: We have modified the text accordingly.

Conclusions

Request 9: lines 450-456 Please explain what do you mean when you write: "Accordingly, reduction or loss of SDHB expression due to mutations or stroma mediated factors during tumor development and progression leads to deficient cell growth control in PDECs and later on PDAC cells thereby contributing to expansion of transformed cells." What are the stoma-mediated factors? Are these the macrophages? And if so, how are the macrophages in pancreatic cancer stroma? Could well be that loss of SDHB is related to a less differentiated or EMT-prone phenotype that might be exploited in cancer, but that would also occur in differentiation and development.

Answer 9: As requested, we have elaborated the conclusion in more detail by providing an outlook on future studies and information on macrophages and further roles of SDHB loss in tumor development and Progression.

Round 2

Reviewer 1 Report

Authors have responded satisfactorily to the reviewer's comments.

please note typo in line 76 (" ")